# Trends of Augmented Reality for Agri-Food Applications

**DOI:** 10.3390/s22218333

**Published:** 2022-10-30

**Authors:** Junhao Xie, Jackey J. K. Chai, Carol O’Sullivan, Jun-Li Xu

**Affiliations:** 1School of Biosystems and Food Engineering, University College Dublin, Belfield, Dublin 4, Ireland; 2School of Computer Science and Statistics, Trinity College Dublin, College Green, Dublin 2, Ireland

**Keywords:** augmented reality, HMD, agriculture, food, nutrition, sensory science

## Abstract

Recent years have witnessed an increasing interest in deploying state-of-the-art augmented reality (AR) head-mounted displays (HMDs) for agri-food applications. The benefits of AR HMDs to agri-food industry stakeholders (e.g., food suppliers, retail/food service) have received growing attention and recognition. AR HMDs enable users to make healthier dietary choices, experience novel changes in their perception of taste, enhance the cooking and food shopping experience, improve productivity at work and enhance the implementation of precision farming. Therefore, although development costs are still high, the case for integration of AR in food chains appears to be compelling. This review will present the most recent developments of AR HMDs for agri-food relevant applications. The summarized applications can be clustered into different themes: (1) dietary and food nutrition assessment; (2) food sensory science; (3) changing the eating environment; (4) retail food chain applications; (5) enhancing the cooking experience; (6) food-related training and learning; and (7) food production and precision farming. Limitations of current practices will be highlighted, along with some proposed applications.

## 1. Introduction

The modern food system that supplies the global population from farm to table has improved over hundreds of years to become a system on a massive scale with increasing sophistication, which involves: growing and harvesting crops; raising and slaughtering livestock; storing, processing and packaging food items; formulation developing; product transporting, distributing and retailing; foodservice, and food preparing at home [1]. Easy access to a variety of nutritious and delicious foods has been enabled through contemporary food science, which integrates multiple disciplines and new technologies. Indeed, many technologies have been utilized within the food system, where the trend is the increasing and wide adoption of new technologies [2,3]. Among these emerging technologies applied in the agri-food sector, augmented reality (AR) technology, in which computer-generated virtual information is superimposed onto the physical world [4], is becoming more and more popular. Although traditionally more widely used in the entertainment and gaming industries, AR technology is now being applied in a wider range of fields [5], as it can provide rich visualizations of information and allow user interaction with the data in context [4,6].

In the agri-food sector, AR technology has delivered many benefits. For example, in terms of a food’s nutrition information, glycemic indices can be visualized and displayed through AR devices, which encourages consumers to be more careful of their food choices, thereby reducing food-related diseases and disorders [7,8]. On the other hand, food can be augmented to look tastier, thereby triggering the desire to eat [9]. Such benefits have received the attention and recognition of stakeholders such as suppliers and producers in the food industry. It is also believed that the use of AR can revolutionize food research methodologies, as AR enables 3D visualization of the internal structures of a food product, making it possible to ‘step inside’ a complex food product. This would enable an innovative evaluation of the internal food structure, which would not be possible with previous approaches [10].

Currently, hand-held AR systems (e.g., smartphones and tablets) are popular and widely available due to affordability; therefore, many researchers have developed AR applications for these devices [11,12,13]. However, these AR systems typically require to be held during their use, which can be troublesome and reduce flexibility. Furthermore, the occurrence of image distortion using the smartphone’s camera can be quite noticeable. On the other hand, head-mounted displays (HMDs: devices worn on the head or as part of a helmet, glasses or visor, which have a small display optic in front of one or each eye), can free up the user’s hands while using the application. Although more expensive than hand-held AR, HMD AR systems are receiving more attention and recognition from researchers and businesses, as evidenced by a significant increase in sales in recent years and future market growth predictions [14].

Several review articles summarize the utilization of AR technology in the agri-food sector. For example, Crofton et al. [10] reviewed AR and MR (Mixed Reality) applications for sensory science and highlighted the potential benefits of AR/MR technology within the food industry. Hurst et al. [15] focused on precision farming and proposed that AR technology for precision farming must be coupled with other technologies, such as mapping algorithms and global positioning systems, in order to fully realize its potential. Rejeb et al. [16] identified enablers of AR in the food supply chain by investigating the relevant AR applications reported. More recently, Chai et al. [17] analyzed AR/MR applications for food and discussed the limitations as well as future research directions of these applications. Nevertheless, most published reviews only focused on a single application topic, for example, sensory science. Meanwhile, there is no review that exclusively focuses on HMDs in the agri-food sector. The objective of this review is to address this research gap by summarizing the latest developments in the use of AR HMDs in the agri-food sector, thereby showcasing the potential benefits of integrating AR with food applications, while also discussing the limitations of these proposed applications.

## 2. Augmented Reality Technologies

AR is broadly defined as a set of technologies that allow computer-generated imagery to be superimposed onto the real world. Calo et al. [18] presented a more detailed definition of AR technology where, in addition to being able to overlay information onto the physical world, it can also: collect various forms of data, recognize and track objects in the physical world, analyze and process data in real-time and provide contextual information. Indeed, these identified features clearly illustrate how an AR device typically works.

In AR applications, markers with distinctive characteristics can be used to easily and accurately track and identify an item using the AR device. However, the main drawback of marker-based AR technology is that only items with markers can be tracked and recognized. The emergence of markerless AR systems allows objects to be identified and tracked by capturing and processing images of those objects. The application of deep learning for object detection and identification using markerless AR technology is reported to be fast as well as accurate [19]. However, the limitation of such approaches is the associated computational burden that might occur.

There are many types of wearable AR devices currently available, among which the most popular include Microsoft HoloLens™, Google Glass™, Epson Moverio™, Vuzix Blade™, Magic Leap™, and GlassUp F4 Smart Glasses™. Such wearable AR devices, also known as optical see-through (OST) heads-up displays, are considered to be the most advanced [19]. These wearables have the advantage of not impeding the user’s vision and at the same time allowing better interactions with the user by facilitating hands-free use.

## 3. Literature Review Methodology

To gather publications pertaining to the use of AR HMDs in the agri-food sector, the keywords ‘augmented reality’, ‘head mounted displays’, ’smart glasses’, ‘AR glasses’, ‘wearables’ ‘food’, ‘agriculture’ and ‘agri-food’ were chosen for an extensive literature search. Combinations of these keywords were then used as input for searches in Google Scholar (https://scholar.google.com/) and Scopus (https://www.scopus.com/). The search was conducted on 15 August 2022, and a time filter was used on both websites to identify related articles published in recent years (i.e., from 2010 to 2022). Relevant publications were initially shortlisted based on the abstracts, results and discussions/summaries, followed by full paper reading to decide on inclusion in the final set of articles to review. The same keyword combinations were also entered in the Google Search engine to find relevant applications that have been developed, reported, commercialized or patented. EndNote 20 software was used for the management of references. After full paper reading, the chosen applications were grouped into seven major categories as summarized in Figure 1. It can be noted that the use of AR HMDs in food production and precision farming, dietary and food nutrition assessment and food sensory science has received relatively more attention compared to other categories in recent years.

## 4. Agri-Food Applications

### 4.1. Dietary and Food Nutrition Assessment

An unhealthy diet is considered to be the culprit of many diseases. For example, it is commonly reported that diets high in energy-dense foods are associated with an increased risk of chronic diseases such as obesity, cardiovascular disease, and specific cancers [20]. Thus, evaluating and assessing foods is necessary to make better dietary decisions. This is especially true for people who need to be careful with their food choices, such as diabetes patients. There are a number of dietary assessment tools available at the moment, e.g., websites providing comprehensive information on a large variety of foods. By consulting those websites, one can discover how many calories are contained in a variety of foods, along with the recommended daily intake amount of a particular food item.

Technological advancements, such as AR HMDs, have enabled the development of better dietary assessment tools. The wearable food nutrition feedback system [21], patented by Microsoft, is a compelling example that illustrates this new way of food evaluation through AR glasses. The system is a transparent HMD with sensors that can detect food items in the user’s field of view. The information is then processed, and feedback such as calorie content and allergens is displayed through the glasses. For example, consider the scenario where a user wearing an AR HMD is about to eat a hamburger. First, the gaze detection elements and the sensors of the system capture and identify the hamburger. Then, relevant nutritional information about the food is retrieved, based on which feedback is generated. A virtual text (i.e., feedback) then appears on the HMD display, which contains a warning that this hamburger may have 800–1200 calories. If the user ignores the information and proceeds to eat the burger, another floating warning appears, stating that it is their second big meal that day and asking if they really want to proceed. This Microsoft patent, an extension of an earlier patent filed in 2012, was filed on 8 October 2015 and granted on 9 May 2017.

Another application for dietary and food nutrition assessment called “Diabetes Shopping”, based on Google Glass, was developed in 2014 for the dietary management of diabetes [7]. In the application’s demonstration video [22], a shopper with diabetes is standing in a grocery store and trying to decide on a vegetable side dish. The shopper picks up corn on the cob and asks the application to determine its glycemic index, which is later displayed by Google Glass. Based on this glycemic index, the shopper can decide whether or not to buy the corn. The Google Glass device is easy to use and provides users with real-time decision support in a private and hands-free way. This important dietary information may help patients with diabetes or family members to make better food choices [7]. However, this application cannot recognize items through the cameras, and needs the user to explicitly interact with the glasses rather than automatically reacting to their actions.

Furthermore, in support of dietary assessment, Fuchs et al. [23] developed a system on the Microsoft HoloLens to detect users’ diet-related activities and display real-time visual interventions to support healthy food choices. The food detection function of this system, based on computer vision and machine learning, achieved high accuracy due to the large number of training images available from publicly available repositories. The detected food item’s nutritional properties and quantity could be found in databases and displayed in real-time for the user. In their experiment, participants were asked to wear a HoloLens headset and stand before a vending machine full of different kinds of snacks and beverages. The HoloLens headset detected the snacks and beverages, and overlaid the user’s view with different color-coded frames corresponding to product-specific Nutri-Score labels. In this way, the user could quickly tell which products were healthy and which were not. The participants in this experiment reported that this system significantly improved their snack and beverage choices, confirming that healthier choices are made when consumers can inspect food information, such as nutritional content and recommended intake, before purchasing and consuming the food. Labelling is an excellent way to inform people about their foods, which can be enhanced even further through the use of AR HMDs. For example, in addition to simply grading foods from “unhealthy” to “healthy”, the HoloLens system by Fuchs et al. [23] also provides visualization of nutritional information. By using Microsoft HoloLens Clicker devices to select a product, detailed nutritional information (Figure 2) for a selected product can be displayed in the user’s view.

Further systems designed to fulfill the function of nutrition visualization and monitoring were identified during this review. Naritomi and Yanai [24] proposed a HoloLens-based system called “CalorieCaptorGlass” that, based on an estimation of the actual size of a food item, calculates its calories using a regression equation. The caloric content calculated is then displayed in the user’s field of view. In this system, size estimation is realized using a deep convolutional neural network. Jiang et al. [8] designed an AR-based wearable food-monitoring system to make food nutritional information quickly available and visible. In their experiment, once food was detected by the Google Glass, its image was uploaded to a server and analyzed. Once the food was recognized, the system accessed an online database maintained by the U.S. Department of Agriculture (USDA) to retrieve its associated nutritional information, which was then returned and displayed on the Google Glass (Figure 3).

In summary, the applications above have demonstrated that, through intuitive visualizations of food properties such as calories, allergens and glycemic index, users can be enabled to perform their own assessment of the foods they saw, and hence better food choices could be made. This suggests that the risk of suffering from a food-related disease may be reduced through the use of AR HMDs, thereby benefiting the users’ health.

### 4.2. Applications in Food Sensory Science

It has been well established in the literature that taste perception is actually a combination of vision, hearing, olfaction, gustation and tactile perception [25,26,27]. Flavors are therefore perceived as tastes based on a variety of stimuli beyond those received by the gustatory system alone. Exploiting this fact, AR HMDs have been deployed to manipulate the perception of food and appetite by changing the appearance and other properties of food. Using an AR HMD and an olfactory display, Narumi et al. [28] transformed a plain cookie into a flavored cookie by overlaying the actual cookie with an image of a flavored cookie (Figure 4). By simultaneously delivering a corresponding scent to the participants with an olfactory display (e.g., a chocolate scent when an image of a chocolate cookie was displayed), many of them were surprised by the great change in taste. An edible marker was used for the AR HMD to detect the actual cookie in this case. In later work, again with an AR HMD, they made a cookie appear bigger in size by superimposing an image of the same cookie of a bigger size. The cookie consumers felt that they had eaten quite a lot and soon achieved satiety, whereas in reality they had eaten only a small amount that was not enough to make them feel full [29]. In terms of food size, Suzuki et al. [30] successfully transformed the perceived volume of a beverage consumed by changing the size of the container of the beverage. In their experiment, participants were asked to wear video see-through (VST) HMDs, while the cups they held were changed into bigger or smaller cups in their field of view. The participants drank significantly greater amounts from a visually bigger cup and significantly smaller amounts from a visually smaller cup. To create the illusion, the relative position of the container was tracked by the camera on the HMD, the image of the container was captured and sent to a laptop for processing, after which the processed image was returned to the HMD and superimposed onto the real container.

Ueda and Okajima [9] altered the appearance of real sushi by superimposing images of a preferred type of sushi. Participants reported that they perceived a better taste while eating the visually changed sushi, although the flavor had not been altered in reality. Ueda’s team also reported that, when the luminance of food was modified in real-time using an AR HMD, perceived taste was improved [31]. Dynamic image processing was used to alter the brightness of food in real-time, by changing the standard deviation of the luminance distribution of food images captured by the HMD. Nakano et al. [32] used generative adversarial network (GAN)-based real-time image-to-image translation in combination with an AR HMD to superimpose curry rice and fried noodles/ramen noodles on plain rice and plain noodles, respectively, which induced changes to participants’ perception of the taste of the food. Narumi et al. [28] presented a system that could visually improve the perceived taste of nutritionally controlled flavorless foods, which has the potential to be used by patients in hospitals. Other approaches, which visually manipulate food/beverage sizes, have been shown to help users to control their food/beverage consumption, thereby improving their health [29,30].

### 4.3. Change the Eating Environment

The environment in which people eat has an influence on their evaluation of their meals, which is an intriguing phenomenon explored by researchers [33,34,35]. Similar to visually changing the food itself, AR HMDs can be used to change the eating environment, thereby enhancing the eating experience. Korsgaard et al. [36] transformed the eating environment into a beautiful virtual park, resulting in an enhanced eating experience as reported by their participants. A similar study was later conducted by the same team, which created a virtual environment using an AR HMD, where participants could enjoy food with virtual avatars while they were eating alone in reality. The results of this study demonstrated that, even though participants were aware that they were eating in a virtual environment, their loneliness was eased [37]. The limitation of Korsgaard’s work was poor blending between the real food and the overlaid virtual environment, which made the synthetic environment more obvious. Nakano et al. [38] developed a method to overcome this limitation and successfully delivered a more immersive experience.

### 4.4. Applications in Food Retail

In order to enhance customers’ experience and increase their willingness to buy, state-of-the-art technologies have been utilized in the food retail industry. One example is the use of electronic digital menus [39] and marker-based AR retail apps [40] in restaurants. AR HMDs have also been used. For example, de Amorim et al. [6] designed an experiment in which participants were asked to put on a HoloLens to shop at a retail food store. When they pointed at a package containing a battered fillet, an overlaid virtual human in a suit appeared, holding the product and speaking about it. It was found that this augmented experience positively influenced participants’ purchase decisions. In an experiment by Lei et al. [41], participants shopped for food products while wearing a Microsoft HoloLens. Using their system, when participants faced product shelves that stored food products, they could choose to amplify, highlight, shrink or hide items to help them with product selection (Figure 5). Amplification was achieved by overlaying the physical product with a virtual image of the product that was 25% bigger; shrinking was the opposite of amplifying (i.e., 25% smaller); highlighting was achieved by overlaying a semi-transparent green color on the physical products; while for covering an opaque grey color was overlaid. The researchers found that their system could speed up the product selection process and suggested that the system could also be used for non-food products. These studies highlight that one of the benefits of using AR HMDs in food retail is that they can enhance the experience of shoppers, thereby potentially increasing their willingness to buy.

### 4.5. Enhancing the Cooking Experience

The potential of using advanced technologies such as food robots and AR HMDs in cooking has also been explored by scientists. Ergün et al. [42] predicted that the global kitchen market will grow rapidly in the future [42]. This has motivated various researchers to explore more efficient and easier-to-use service operations, with the goal of realizing remote control, time-saving and cost-saving in cooking processes. In an interesting study by Ergün et al. [42], an “AR-Supported Induction (AR-SI) cooker” was developed for the Microsoft HoloLens HMD, which provided supportive experiences for the preparation of food. While preparing a meal and wearing the HMD, cooking instructions were displayed in the participant’s field of view. Moreover, the participant could hear the instructions at the same time through the built-in speakers of the HoloLens. It was claimed that, with their well-designed AR-SI cooker, a highly complex meal could be accurately and quickly created by someone with no cooking experience. A similar application called Giga AR Cooking, based on Nreal AR glasses, has been commercialized in Germany by Vodafone and Nreal since March 2021 [43]. With this application, users can see overlays of cooking instructions displayed through the glasses, accompanied by an animated virtual avatar of the famous German chef, Steffen Henssler, sharing his cooking tips and tricks.

### 4.6. Food-Related Training and Learning

Augmented Reality is a well-established technology that supports different industrial needs and is one of the leading technologies in learning and training processes [44]. When processes are augmented and thereby clarified through the use of AR devices, trainees and learners are more motivated [45], resulting in reduced worker hours and costs [46]. As reported by Clark et al. [47], the use of AR HMDs brings innovation and effectiveness to food safety training in the food industry. One such application was developed by Albayrak et al. [48], who used AR HMDs to effectively train restaurant workers using gamification and personalization. They demonstrated that using their AR application could shorten training times and increase workers’ willingness to learn, thereby improving the quality of service. Furthermore, the time invested by the restaurant management on one-to-one tutorials could be reduced, or even eliminated, resulting in a financial benefit to the restaurant.

In a study conducted by Christensen and Engell-Nørregård [49], slaughterhouse employees were assisted by AR HMDs in trimming pig bellies. Based on a computerized tomography (CT) scan of the belly, an image showing the thickness of the underlying fat in different colors was generated and overlaid, via the HMD, on the actual pork belly. Participants wearing the HMD could follow overlaid instruction text (Figure 6A) while observing the distribution and thickness of fat (Figure 6B). These rich visualizations helped them to trim the pork more efficiently. For the purposes of teaching unskilled workers in the meat processing sector, Dhiman et al. [50] deployed intelligent assistants that harnessed expert skills and knowledge, along with various sensors, to detect the actions of trimming, separation and deboning of specific parts of a pork shoulder. Real-time instructions and feedback on the detected actions were then displayed via the HMD. These studies all serve to demonstrate the great benefits that can be gained from using AR HMDs to improve the effectiveness of food-related training and learning.

### 4.7. Food Production and Precision Farming

Developed since the 1990s, precision farming is defined as “the use of information technologies for the assessment of physical resource variability aimed at improved management strategies for optimizing economic, social and environmental farming” [51]. In other words, farmers are enabled to optimize and increase productivity by putting in place a series of targeted key interventions, using advanced digital strategies and tools. Precision farming is an inherently interdisciplinary field incorporating concepts from bioinformatics, biostatistics, grass and animal breeding, animal husbandry and nutrition, machine learning, sensor networking, autonomic network management and engineering. AR glasses and HMDs have recently been proposed to improve the implementation of precision farming. For example, Santana-Fernández et al. [52] designed a GPS guidance system for agricultural tractors using an AR HMD, which allowed a farmer driving a tractor to see treated and untreated areas of farmland clearly. Green overlays were displayed on the treated areas through the AR glasses, while the untreated areas were not overlaid with anything.

Soil sampling is essential in agriculture, as it allows farmers to know which zones of their land are good for crop growth (i.e., rich in nutrients) and which areas need to be improved. Soil sampling also allows farmers to make proper decisions on the use of fertilizers (e.g., type and amount). Huuskonen and Oksanen [53] describe an approach that uses AR glasses in combination with unmanned aerial vehicles (UAV) in order to increase the accuracy of soil sampling. After ploughing, a soil map is produced from UAV imaging and an algorithm determines the locations for soil sampling. Then, a sample collector is guided by the AR glasses to the precise sampling locations.

Most fruits, including tomatoes, do not ripen at the same time, even if they belong to the same plant. If a worker lacks experience in harvesting tomatoes, they may not be able to accurately tell whether the fruits are ready for harvest or not. Therefore, a tomato harvest support system [54] has been proposed, which uses a Microsoft HoloLens, a wireless connection and a computer to process data. When tomato fruits are captured by the HoloLens, the images of the fruits are first sent to the computer for data processing, where sugar content and acidity are calculated. The HoloLens then displays an enclosing virtual frame for each fruit, labeled with the corresponding sugar content and acidity values (Figure 7). Sugar content and acidity values indicate ripeness and are calculated by the computer based on the colors of the captured fruit.

Xi et al. [55] developed a system to support water quality inspection in the prawn farming industry. Participants in their experiment wore an AR HMD, through which they could visualize information about the water quality of ponds in a prawn farm and quickly locate those ponds in which water quality was compromised. Finally, increasing productivity in livestock farming was the goal of Caria et al. [56], who developed an application on GlassUp F4 Smart Glasses for AR. A user wearing the glasses can scan QR codes attached to animals by simply looking at them. Information about the animals stored in a database is then retrieved and displayed as augmented features in the user’s field of view, thus enabling them to access data quickly and in a hands-free manner. In conclusion, all of the previous case studies reviewed serve to demonstrate how Augmented Reality can improve the implementation and deployment of precision farming applications.

## 5. Limitations and Future Research Directions

AR HMDs have some drawbacks currently that can limit their more widespread use. For example, a live (wireless) connection is needed in most cases for the transfer and transformation of data, while the speed of the connection is crucial to achieving real-time performance. The limited battery capacity and computing power of AR wearables can also be a limiting factor [57]. Precise food item recognition and localization using AR devices in an uncontrolled environment are still relatively hard to achieve in practice [58]. Although AR devices can easily recognize markers attached to items, the use of markers restricts augmentation to only those items that are marked. Furthermore, when a marked item is too far away from the AR device, detection accuracy is reduced. Research and development of markerless AR technologies has allowed some progress to be made, but high illumination levels of the target objects/areas and a clear background (e.g., pure white) are needed to increase the chances of precise recognition. In real scenes, however, food items are usually not well-lit and the background is often complex. The chances of a true positive recognition are increased when foods are not closely grouped together, but in real-time scenarios, food products are usually aggregated. Furthermore, when part of a food is occluded or missing (e.g., when part of a cookie is eaten [28]), or its orientation is different from its picture in the database, it is unlikely to be accurately recognized.

Even if a food item is accurately recognized by an AR device, the information about the food item returned via the AR device is not necessarily accurate. The main reason for this drawback is that the AR system accesses information about the items mainly based on their visual properties, but food items with similar visual properties do not necessarily share the same nutritional attributes. For example, AR glasses could identify a hamburger based on its typical appearance but, if it cannot detect whether this hamburger is made from chicken or beef, the retrieval of accurate nutritional information would be impossible. In terms of ethical issues, the ability of AR systems to record videos, take photos and even identify faces [59] “secretly” [60] poses a threat to the privacy of others [61]. In addition, the user’s own privacy may be jeopardized, as they could be monitored by unauthorized parties through the AR device [62]. Last but not least, the relatively high costs of this technology is an obstacle to more widespread use of AR HMDs [17], although increasing demand may drive prices down in the future.

An interesting direction for future work would be to combine AR wearables with current state-of-the-art techniques, such as hyperspectral imaging, to promote the development of new research and practical applications for the agri-food sector. Furthermore, AR applications could be developed to visualize important properties of food, such as traceability or halal status, for which there are currently no relevant AR HMD applications available.

## 6. Conclusions

This timely review focuses on applications of AR HMDs in the agri-food sector and will help to promote an understanding of AR technology and the latest developments in this field. As such, it should be of interest to many readers from a broad range of related communities. The highlighted benefits of deploying AR wearables include: enabling healthier food choices, improving the enjoyment of food consumption, making cooking easier, facilitating more effective learning of food-related knowledge, implementing better precision farming systems, and increasing productivity in food production. Consequently, researchers have been motivated to develop new applications to realize the unique fusion of food and AR, combining knowledge from diverse fields such as food science, chemistry, biotechnology, psychology, computer graphics, and computer vision. It is anticipated that, despite the discussed obstacles that are impeding more widespread adoption, the development of AR HMDs for the agri-food sector will continue to evolve in the future.

## Figures and Tables

**Figure 1 sensors-22-08333-f001:**
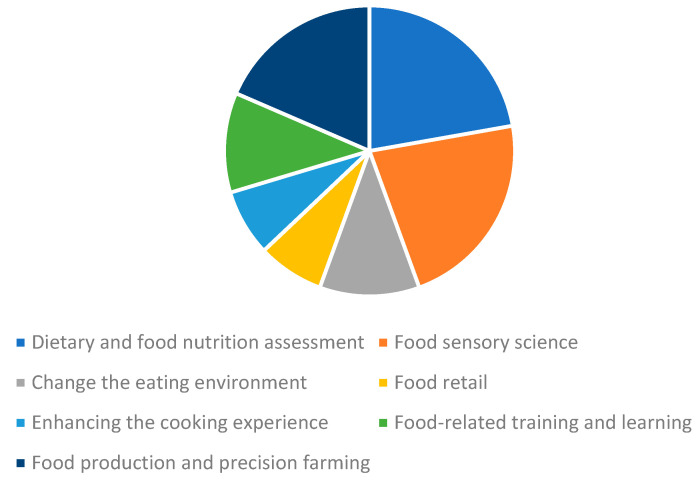
The proportions of AR HMDs used in the seven application categories.

**Figure 2 sensors-22-08333-f002:**
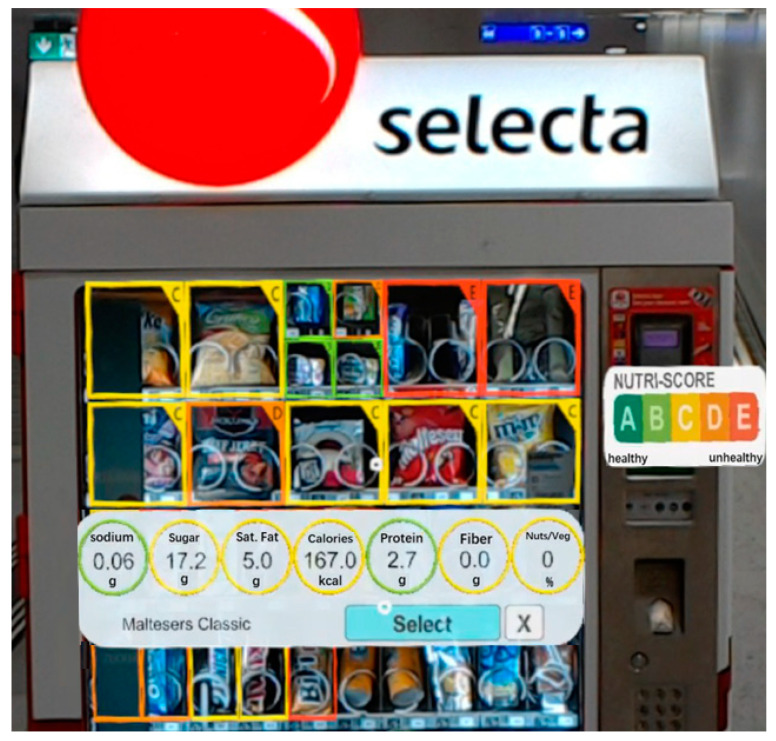
Detailed nutrition information of the selected snack displayed by the HoloLens. Reproduced from Fuchs et al. [23].

**Figure 3 sensors-22-08333-f003:**
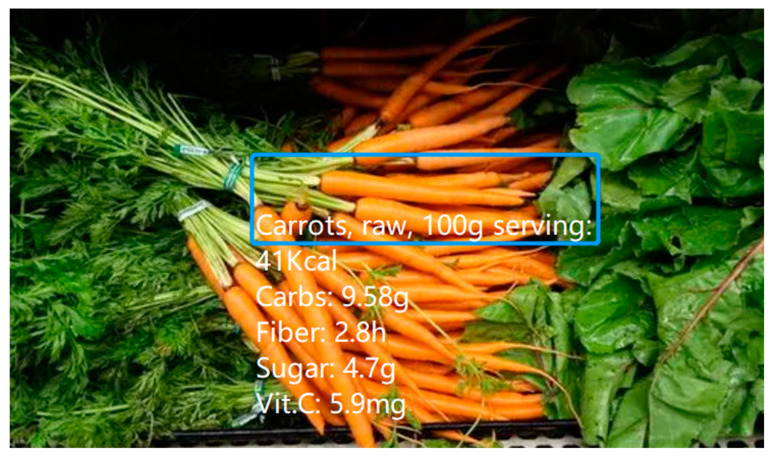
The nutritional information on a tracked grocery item. The blue box represents the location of the food as tracked by the proposed system. Information for several nutrients is reported per 100 g. Reproduced from Jiang et al. [8].

**Figure 4 sensors-22-08333-f004:**
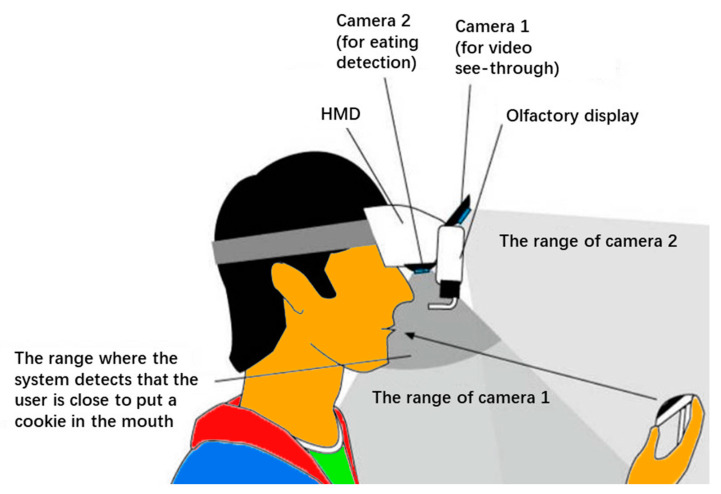
Layout of two cameras, a head-mounted display and an olfactory display. Adapted with permission from Ref. [28]. 2016, Association for Computing Machinery.

**Figure 5 sensors-22-08333-f005:**
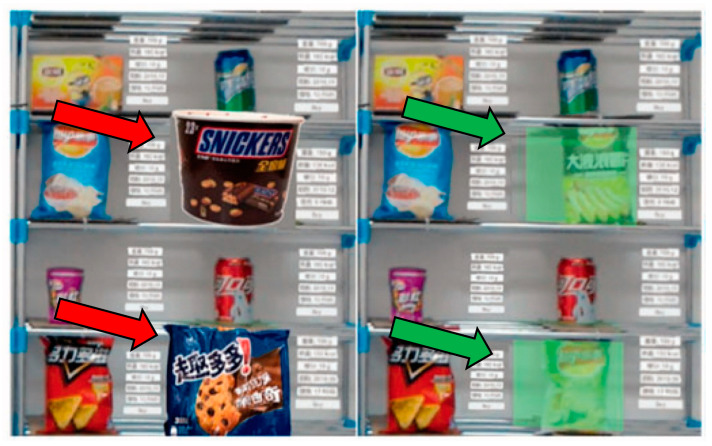
Amplified products are indicated by red arrows (**left**) and highlighted products are indicated by green arrows (**right**). Adapted with permission from [41]. 2022, Taylor & Francis.

**Figure 6 sensors-22-08333-f006:**
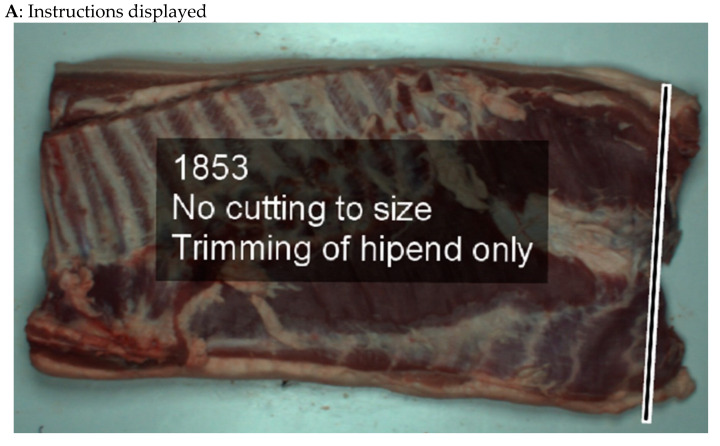
Augmented information includes visualized instructions on how to cut the belly (**A**) and color-coded fat layer thicknesses (**B**). Reproduced from Christensen and Engell-Nørregård [49]. In (**B**), blue indicates the thickness of fat below 7 mm, green to orange indicate thickness between 7 mm and 15 mm and red indicates thickness above 15 mm.

**Figure 7 sensors-22-08333-f007:**
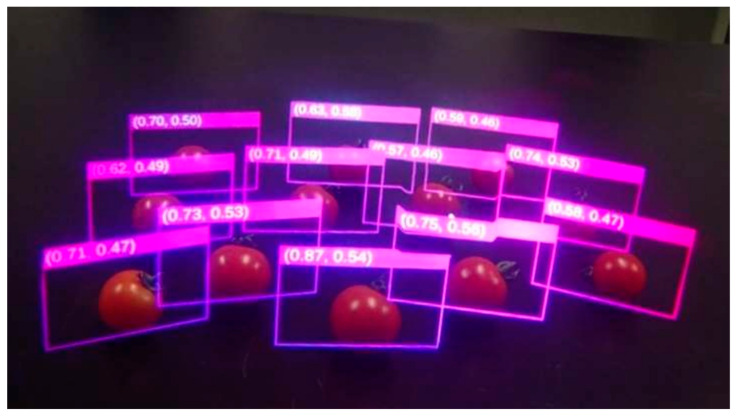
Acidity and sugar content returned by the system. Adapted with permission from [54]. 2022, Institute of Electrical and Electronics Engineers.

## Data Availability

Not applicable.

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
