# Peer review of "Trends of Augmented Reality for Agri-Food Applications"

_sensors, 2022, doi:10.3390/s22218333_

Round 1

Reviewer 1 Report

This paper reviewed the current applications of AR and HMDs in the agri-food industry. Seven categories of applications were introduced to highlight the existing work. The limitations of AR and HMDs were also discussed.

Overall, this paper contributed to the agri-food and AR research field and fostered future research in this direction. However, the objective of the review is not clear in the paper. The authors only introduced the background of the agri-food sector and the emerging AR technology but did not indicate the purpose of this review. In addition, the research methodology is not provided in the manuscript. The authors should provide information such as how they collected the literature or how the seven categories were defined. Finally, it is suggested to include a table or a figure to summarize the literature in each category.

Author Response

Reviewer #1: Review of the Manuscript, comment 1.

"The objective of the review is not clear in the paper. The authors only introduced the background of the agri-food sector and the emerging AR technology but did not indicate the purpose of this review".

Author response: We thank the reviewer for investing his/her time on reading this manuscript and providing invaluable suggestions. Following the feedback, the objective has been added as the final paragraph of the Introduction section, please see below:

“Several review articles summarize the utilization of AR technology in the agri-food sector. For example, Crofton et al. [10] reviewed AR and MR (Mixed Reality) applications for sensory science and highlighted the potential benefits of AR/MR technology within the food industry. Hurst et al. [15] focused on precision farming and proposed that AR technology for precision farming must be coupled with other technologies, such as mapping algorithms and global positioning systems, in order to fully realize its potential. Rejeb et al. [16] identified enablers of AR in the food supply chain by investigating the relevant AR applications reported. More recently, Chai et al. [17] analyzed AR/MR applications for food and discussed the limitations as well as future research directions of these applications. Nevertheless, most published reviews only focused on a single application topic, for example, sensory science. Meanwhile, there is no review that exclusively focuses on HMDs in the agri-food sector. The objective of this review is to address this research gap by summarizing the latest developments in the use of AR HMDs in the agri-food sector, thereby showcasing the potential benefits of integrating AR with food applications, while also discussing the limitations of these proposed applications.

Reviewer #1: Review of the Manuscript, comment 2.

" In addition, the research methodology is not provided in the manuscript. The authors should provide information such as how they collected the literature or how the seven categories were defined. Finally, it is suggested to include a table or a figure to summarize the literature in each category.".

Author response: We thank the reviewer for pointing this out. Indeed, literature review methodology is of great importance to a review paper. In this regard, we added a section (Section 3 of the revised manuscript) to introduce how we find relevant papers and how we defined the categories of the agri-food sectors. A graph (figure1) has been made to summarize the literature in each application category. Please see below.

“3. Literature review methodology

To gather publications pertaining to the use of AR HMDs in the agri-food sector, the keywords ‘augmented reality’, ‘head mounted displays’, ’smart glasses’, ‘AR glasses’, ‘wearables’ ‘food’, ‘agriculture’ and ‘agri-food’ were chosen for an extensive literature search. Combinations of these keywords were then used as input for searches in Google Scholar (https://scholar.google.com/) and Scopus (https://www.scopus.com/). The searches were conducted in August 2022, and a time filter was used on both websites to identify related articles published in recent years (i.e., from 2010 to 2022). Relevant publications were initially shortlisted based on the abstracts, results and discussions/summaries, followed by full paper reading to decide on inclusion in the final set of articles to review. The same keyword combinations were also entered in the Google Search engine to find relevant applications that have been developed, reported, commercialized or patented. EndNote 20 software was used for the management of references. After full paper reading, the chosen applications were grouped into seven major categories as summarized in Fig. 1. It can be noted that the use of AR HMDs in food production and precision farming, dietary and food nutrition assessment and food sensory science has received relatively more attention compared to other categories in recent years.

Reviewer 2 Report

- In the abstract, the authors mention the benefits being great, but what are these benefits? These benefits need further elaboration than what is in the second paragraph of the introduction.

- The citation format is not author, year. Pease follow the journal guidelines for the citation and references.

- More insight into the different application would be appreciated.

- Research directions and suggested applications would be a plus for the paper. 

- In a short note, what is the benefit of this paper to the readership?

- What other similar reviews are available and what are their limitations.

Author Response

Reviewer #2: Review of the Manuscript, comment 1.

" - In the abstract, the authors mention the benefits being great, but what are these benefits? These benefits need further elaboration than what is in the second paragraph of the introduction.".

Author response: We thank the reviewer for pointing this out. According to the suggestion, the benefit of AR in the agri-food application has been added in the abstract, as below:

AR HMDs enable users to make healthier dietary choices, experience novel changes in their perception of taste, enhance the cooking and food shopping experience, improve productivity at work and enhance the implementation of precision farming.”

In addition to the second paragraph of the introduction, the benefits for each application theme are further elaborated in Section 4.

Reviewer #2: Review of the Manuscript, comment 2.

" - The citation format is not author, year. Pease follow the journal guidelines for the citation and references.".

Author response: We thank the reviewer for pointing this out. We have downloaded the recommended Reference Style file from the journal guidelines (https://www.mdpi.com/journal/sensors/instructions) and applied the style in the manuscript using EndNote.

Reviewer #2: Review of the Manuscript, comment 3.

" - More insight into the different application would be appreciated.".

Author response: We thank the reviewer for giving this advice. For that, we have added details to every application category. These newly added details could be found in section 4 and section 5 of the track-change version of the manuscript.

Reviewer #2: Review of the Manuscript, comment 4.

" Research directions and suggested applications would be a plus for the paper".

Author response: We thank the reviewer for giving this advice. We have added another paragraph in Section 5 to discuss the future research direction, please see below:

“An interesting direction for future work would be to combine AR wearables with current state-of-the-art techniques, such as hyperspectral imaging, to promote the de-velopment of new research and practical applications for the agri-food sector. Fur-thermore, AR applications could be developed to visualize important properties of food, such as traceability or halal status, for which there are currently no relevant AR HMD applications available.”

Reviewer #2: Review of the Manuscript, comment 5.

" In a short note, what is the benefit of this paper to the readership?".

Author response: We thank the reviewer for giving this advice. We have inserted a sentence in the Conclusion section:

“This timely review focuses on applications of AR HMDs in the agri-food sector and will help to promote an understanding of AR technology and the latest developments in this field. As such, it should be of interest to many readers from a broad range of related communities.”   

Reviewer #2: Review of the Manuscript, comment 6.

" - What other similar reviews are available and what are their limitations.".

Author response: We thank the reviewer for suggesting this. Similar reviews do exist in the literature. We added a paragraph to the introduction section to briefly discuss similar reviews and their limitations, please see below:

Several review articles summarize the utilization of AR technology in the agri-food sector. For example, Crofton et al. [10] reviewed AR and MR (Mixed Reality) applications for sensory science and highlighted the potential benefits of AR/MR technology within the food industry. Hurst et al. [15] focused on precision farming and proposed that AR technology for precision farming must be coupled with other technologies, such as mapping algorithms and global positioning systems, in order to fully realize its potential. Rejeb et al. [16] identified enablers of AR in the food supply chain by investigating the relevant AR applications reported. More recently, Chai et al. [17] analyzed AR/MR applications for food and discussed the limitations as well as future research directions of these applications. Nevertheless, most published reviews only focused on a single application topic, for example, sensory science. Meanwhile, there is no review that exclusively focuses on HMDs in the agri-food sector. The objective of this review is to address this research gap by summarizing the latest developments in the use of AR HMDs in the agri-food sector, thereby showcasing the potential benefits of integrating AR with food applications, while also discussing the limitations of these proposed applications.”

Round 2

Reviewer 1 Report

The authors have addressed all concerns the reviewer had. Please conduct a proofread to correct some minor issues.

Reviewer 2 Report

The authors addressed my concerns.